Cbl upregulates cysH for hydrogen sulfide production in Aeromonas veronii

Zhang Yidong
Liu Zebin
Tang Yanqiong
Ma Xiang
Tang Hongqian
Li Hong
Liu Zhu zhuliu@hainanu.edu.cn
Hainan University , Haikou , China
Brazelton William
Electronic publication date: 2021 Sep 9
Publication date: 2021
Volume: 9
Electronic Location ID: e12058
Received 2021 Mar 30; Accepted 2021 Aug 4
Copyright: © 2021 Zhang et al.
Copyright year: 2021
Copyright holder: Zhang et al.
License: This is an open access article distributed under the terms of the Creative Commons Attribution License, which permits unrestricted use, distribution, reproduction and adaptation in any medium and for any purpose provided that it is properly attributed. For attribution, the original author(s), title, publication source (PeerJ) and either DOI or URL of the article must be cited.
License URL: https://creativecommons.org/licenses/by/4.0/

Keywords: SmpB, Cbl, H2S metabolism, Regulation

Funding: National Natural Science Foundation of China 31772887 This work was supported by the grants from the National Natural Science Foundation of China Nos. 31772887 (to Zhu Liu). The funders had no role in study design, data collection and analysis, decision to publish, or preparation of the manuscript.

==============================
Endogenous hydrogen sulfide (H2S) is generated in many metabolism pathways, and has been recognized as a second messenger against antibiotics and reactive oxygen species (ROS). In Aeromonas veronii, Small Protein B (SmpB) plays an important role in resisting stress. The absence of smpB could trigger sulfate assimilation pathway to adapt the nutrient deficiency, of which was mediated by up-regulation of cbl and cys genes and followed with enhancing H2S production. To figure out the mutual regulations of cbl and cys genes, a series of experiments were performed. Compared with the wild type, cysH was down-regulated significantly in cbl deletion by qRT-PCR. The fluorescence analysis further manifested that Cbl had a positive regulatory effect on the promoter of cysJIH. Bacterial one-hybrid analysis and electrophoretic mobility shift assay (EMSA) verified that Cbl bound with the promoter of cysJIH. Collectively, the tolerance to adversity could be maintained by the production of H2S when SmpB was malfunctioned, of which the activity of cysJIH promoter was positively regulated by upstream Cbl protein. The outcomes also suggested the enormous potentials of Aeromonas veronii in environmental adaptability.

Introduction

Aeromonas veronii is widely present in fresh water, sewage, soil and even sea water (Hickman-Brenner et al., 1987), which endows with strong resistance to multiple antibiotics (Liu et al., 2018). Small Protein B (SmpB) acts as a small RNA binding protein in the trans-translation system to help transfer messenger RNA (tmRNA) to rescue the retained ribosomes in bacteria (Wali Karzai, Susskind & Sauer, 1999). Also, SmpB performs many significant functions in biological regulation. For example, the expression of ribonuclease R (RNase R), an exonuclease molecule that recognizes and degrades RNA, depends on SmpB in Streptococcus pneumoniae (Moreira et al., 2012). SmpB protein promotes the binding and degradation of RNase R by HslUV and Lon in Escherichia coli (Liang & Deutscher, 2012). Moreover, SmpB has similar effects with the known RNA chaperone proteins such as CrsA and Hfq. The loss of SmpB affects 4% transcription changes of genes in salmonella, of which involves in biological processes, including invasion, bacterial movement, central metabolism, lipopolysaccharide (LPS) biosynthesis, two-component regulatory system and fatty acid metabolism (Sittka et al., 2007; Ansong et al., 2009). In all, SmpB is essential for intra-macrophage proliferation and the strong adaptability to oxidative stress in Salmonella (Ansong et al., 2009). However, the mechanism of SmpB increasing the adaptability to stress is vague. To survive in oxidative damage, the bacteria evolve the enzymatic antioxidants such as superoxide dismutase (SOD) to tolerate peroxide anion radicals (He et al., 2017). Furthermore, bacterial hydrogen sulfide (H2S) could increase SOD activity and maintain redox balance in vivo (Shukla et al., 2017). H2S is a gaseous molecule with an unpleasant smell which at high concentrations is toxic to most living organisms/live cells, etc. (Lindenmann et al., 2010). However, the low concentration of H2S participates in bacterial defense against reactive oxygen species (ROS) and antibiotics-induced oxidative damage (Lindenmann et al., 2010). Since H2S is water-soluble, a considerable amount of H2S exists in the form of HS−1. It is hardly distinguished whether H2S or HS−1 that contributes to the biological activity. Therefore, H2S mentioned in biologically relevant article virtually contains both species in case of confusion (Powell, Dillon & Matson, 2018). One of the basic H2S synthesis includes the sulfate assimilation pathway, which is catalyzed by cysNDC and cysJIH (Shatalin et al., 2011; Kimura, 2014; Wu et al., 2015). In Salmonella typhimurium, the expression of cysJIH is regulated by CysB which has 41% amino acid sequence homology with Cbl (Iwanicka-Nowicka et al., 2007; Álvarez et al., 2015), there is little evidence regarding the role of Cbl in regulating cysJIH in Aeromonas species. Both CysB and Cbl are LysR-type transcriptional activator. In sulfur metabolism, Cbl acts as a sensor of the intracellular sulphate level, and activates tau and ssu promoter in vivo and in vitro (Van Der Ploeg et al., 1999; Van Der Ploeg, Eichhorn & Leisinger, 2001; Bykowski et al., 2002). In addition, Cbl activates sulfate starvation-induced genes under sulfate starvation (Van Der Ploeg et al., 1999). Taking together, there may be a potential connection between Cbl and cys genes in the sulfate assimilation pathway.

In Aeromonas veronii, both SmpB and H2S play important roles in adverse stress. The absence of SmpB induced the generation of H2S helping to survive. The transcriptomic analysis revealed that both cbl and cys genes were up-regulated in SmpB deletion strain. To clarify the regulatory relationship between cbl and cys genes in the sulfate assimilation pathway, real-time PCR experiment and fluorescence analysis were performed, showing that Cbl positively regulated cysH gene. Furthermore, bacterial one-hybrid system and EMSA verified that Cbl regulated cysH by binding to the promoter of cysJIH. In brief, Cbl bound and activated cysJIH promoter directly to increase H2S production, remedying the survival ability after smpB deficiency. Our study elucidated the strong vitality and adaptability of Aeromonas veronii in adverse stress. We also uncovered a novel model of H2S biosynthetic mechanism that may be contributed to stressful survival and recalcitrance of bacterial infections.

Materials & methods

Bacterial strains, plasmids and culture conditions

The bacterial strains and plasmids used in this study were shown in Table S1. The smpB deletion strain of Aeromonas veronii C4 was constructed previously (Liu et al., 2015). The derivative Aeromonas veronii C4 strains were grown in LB/M9 medium supplemented with 50 mg/mL ampicillin at 30 °C, and E. coli strains were grown in LB medium supplemented with 50 mg/mL kanamycin and 25 mg/mL chloramphenicol at 37 °C. And all plasmids were sequenced for verification. LB medium contained 10% tryptone, 5% yeast extract and 10% NaCl. M9 medium contained 20% 5×M9 salts, 0.2% 1 M MgSO4, 0.01% 1 M CaCl2 and 0.4% Glucose, of which 5×M9 salts included 6.4% Na2PO4·7H2O, 1.5% KH2PO4, 0.25% NaCl and 0.5% NH4Cl.

Complemented strain construction

The DNA fragment including both cbl gene and its promoter was amplified by PCR (Wang et al., 2019). PCR reaction was performed as follow: 98 °C for 2 min, stepped by 98 °C for 30 s, 55 °C for 30 s and 72 °C for one kb/min in 30 cycles. The purified PCR product was inserted into pBBR plasmid for creating pBBR-Cbl expression plasmid. The cbl deletion strains were complemented by the conjugation of recipient WM3064 strains carrying pBBR-Cbl.

H2S detection

The Pb(Ac)2 detection (Shatalin et al., 2011) method and WSP5 fluorescent H2S probe (Peng et al., 2014) were performed for monitoring H2S production in gas and liquid phases, respectively. Bacteria were grown in M9 at 30 °C for 48 h with soaked Pb(Ac)2 paper strips hanging from the mouth of the conical bottle, and five mg/L Na2SO3 was added as a source of sulfur. Pb(Ac)2-soaked paper strips showed a PbS brown stain as a result of the reaction with H2S. The color length of one mm represented 12 μg/L of H2S production. After 10 μM WSP5 was added to liquid bacterial culture, the samples were incubated at 37 °C for 30 min and then washed in PBS buffer to remove excess probe. Synergy H1 (BioTek) was used to take fluorescent readings at excitation 500 nm and emission 533 nm. Each reaction was performed at least in triplicate.

RNA extraction and qRT-PCR

The qPCR reaction was conducted with ABI Prism® 7300 (ABI, New York, NY, USA) for fluorescent detection utilizing SYBRR® Green real time PCR Master Mix (Toyonbo, Shanghai, China). The cDNA was synthesized by RNA reverse transcription reaction and was used as the template for real-time PCR. The primers used to monitor expression of the objective genes were summarized in Table S2. Each reaction was performed at least in triplicate, and wild type (WT) was chosen as the control. And the data was analyzed by the comparative CT method (Schmittgen & Livak, 2008).

Fluorescence analysis

The promoter of cysJIH was fused with eGFP and inserted into pUC19 plasmid. The cbl gene was cloned into pTRG plasmid simultaneously. Both the above plasmids were co-transformed into E. coli Reporter strain. Meanwhile, the recombinant pUC19 plasmid and the empty pTRG plasmid were co-transformed as the negative control. After bacteria were grown in LB at 37 °C, the total amount of 1 × 108 cells were harvested in 1.5 mL eppendorf tube at interval time. The samples were washed with PBS twice, and placed on Synergy H1 (Biotek) for the fluorescent readings at excitation 425 nm and emission 525 nm. Each reaction was performed at least in triplicate.

Bacterial one-hybrid analysis

To identify whether the transcription factor Cbl interacted with the promoter of cysJIH, Cbl was inserted into pTRG, and the promoter of cysJIH was ligated with pBXcmT, following with both the recombinant plasmids were cotransformed into E. coli Reporter strain. The transformants were placed on a selective NM medium plate containing five mM 3-amino-1, 2, 4-triazole (3-AT) and 12.5 mg/mL streptomycin for incubation at 37 °C for 48 h. The pTRG and pBXcmT plasmids were co-transformed as the negative control, and pTRG-GAL and pBT-LGF2 were co-transformed as the positive control.

Protein expression and purification

The cbl gene was inserted into pET28a plasmid and transformed into E. coli BL21 strain. The expression and purification were performed according to previous procedure (Bykowski et al., 2002). Cbl protein was purified from E. coli BL21 harboring the pET28a-Cbl plasmid. The recombinant bacteria were grown in LB at 37 °C until the logarithmic phase, followed by the addition of 0.1 mM IPTG to induce protein expression at 15 °C for 14 h. The cells were harvested in Tris-HCl and lysed by sonication. The supernatant was collected after centrifugation and loaded onto a Ni-NTA column. The sample was eluted prior to the dialysis, and SDS-PAGE was used to assess protein purity.

Electrophoretic mobility shift assay (EMSA)

Double stranded DNA probes were radiolabeled with Fluorophore 6-carboxy-fluorescein (FAM) and purified by FastPure Gel DNA Extraction Mini Kit (Vazyme). For the EMSA, DNA probe was incubated with Cbl protein samples in reaction buffer (10 mM Tris–HCl, one mM MgCl2, one mM DTT, 40 mM KCl, 0.1 mg/mL BSA, 5% (w/v) glycerol) at 37 °C for 30 min. After the samples were separated using a 6% native acrylamide gel (Zhang et al., 2020), the gel was then exposed to a phosphorscreen and visualized on Typhoon FLA 9500.

Transcriptome analysis

To perform the whole-transcriptome analysis, the wild type and smpB deletion of Aeromonas veronii C4 were grown in M9 at 30 °C for 20 h, and 2 OD600 of cells were collected. Illumina HiSeq-X ten based on the service of RNA-Seq Quantification library at BGI-Shenzhen (China) was used to obtain the transcriptome sequencings. And the RNA-seq raw data was assembled and analyzed by comparing with the translational region of the annotated DNA sequence in reference genome (GCA_001593245.1) using HISAT (Kim, Langmead & Salzberg, 2015). The DESeq. 2 package in R was used for the estimation of fold changes and other analysis (Love, Huber & Anders, 2014).

Statistical analyses

Statistical significance was determined by t test (two-tailed distribution with two-sample, equal variance) when directly comparing two conditions or a one-way analysis of variance (ANOVA) and Tukey post-test by pairwise comparisons.

Results

Transcriptomic analysis

Based on the transcriptomic analysis, the deletion of SmpB mainly caused the changes in 20 biological pathways, including two-component system, sulfur metabolism, plant pathogenic bacteria interaction, and phenylalanine metabolism. Sulfur metabolism was the most influential on metabolic pathways (Fig. 1A).

Figure 1 Transcriptomic analysis between wild type (WT) and smpB knockout.

(A) The KEGG pathways for the different metabolites between WT and the smpB deletion (ΔsmpB). (B) The relative expression of the correlated H2S synthesis genes in WT and ΔsmpB cells. Values represented means ± SD (n = 3). ***p < 0.001 was determined by one-way ANOVA and Tukey post-test. (C) The deletion of smpB enhanced the expression of genes in the sulfate assimilation pathway.

In Aeromonas veronii C4, the H2S synthesis pathway included the sulfate assimilation pathway, the organic pathway, and the 3-MST pathway. But compared with others, Aeromonas veronii C4 lacked cystathionine β-synthase (CBS) in the transsulfuration pathway and cysteine aminotranferase (CAT) in the 3MST pathway. The deletion of SmpB mainly up-regulated the transcription levels of cysN, cysD, cysC, cysH, cysJ, cysI and cbl (Fig. 1B). Also, these genes were mainly involved in sulfate assimilation pathway (Fig. 1C). The transcription of cysB did not change. Therefore, we speculated that SmpB deficiency was able to increase H2S synthesis.

The production of H2S was increased in the absence of SmpB under nutritionally deficient conditions

To figure out how sulfur metabolism was affected by SmpB deficiency, H2S production was measured in rich and deficient nutrition conditions by Pb(Ac)2 detection test. There is no difference between wild type (WT) and smpB deletion in a rich medium (LB medium) (Fig. 2A). Under the condition of nutritional deficiency (M9 medium), the smpB deletion produced less amount of H2S in the early stage of growth, but it enhanced to synthesize H2S in the stationary stage (Fig. 2B). The final H2S yield of smpB deletion was significantly higher than that of WT. This suggested that the production of H2S was increased in the absence of SmpB during auxotrophic conditions, especially predominant during the stationary phase of bacterial growth.

Figure 2 The production of H2S were increased in the absence of SmpB under nutritionally deficient conditions.

(A) H2S production was measured by Pb(Ac)2-soaked paper strips in LB medium supplemented with five mg/L Na2SO3. No significant differences existed between WT and ΔsmpB strains. (B) H2S production was measured by Pb(Ac)2-soaked paper strips in M9 medium supplemented with five mg/L Na2SO3. Values represented as means ± SD (n = 3). **p < 0.005 was determined by one-way ANOVA and Tukey post-test.

Cbl affects the generation of H2S

Using both the classic Pb(Ac)2 detection test and a fluorescent-based probe WSP5 (Zhang et al., 2020), we confirmed that, the production of H2S in cbl deletion strain was significantly lower than that of WT in M9 medium (Figs. 3A and 3B). And the difference was offset when Cbl protein was complemented (Figs. 3A and 3B). All the results were consistent with the transcriptome data, implying that Cbl had a positive regulatory effect on the synthesis of H2S under nutritional deficiency.

Figure 3 Cbl affected H2S production by promoting the transcription of cysH.

(A) H2S production was measured by Pb(Ac)2-soaked paper strips in M9 medium supplemented with five mg/L Na2SO3. The tested strains included WT, Δcbl and the complemented strain (Δcbl-Pcbl). Values represented as means ± SD (n = 3). **p < 0.005 was determined by one-way ANOVA and Tukey post-test. (B) Fluorescence intensities were detected by Synergy H1 (BioTek, Winooski, VT, USA) after the tested strains were treated with fluorescent H2S probe in M9 medium. Values represented as means ± SD (n = 3). *p < 0.01 was determined by one-way ANOVA and Tukey post-test. (C) The relative expressions of H2S synthesis genes were detected by qRT-PCR. Values represented as means ± SD (n = 3). *p < 0.01 was determined by one-way ANOVA and Tukey post-test. (D) Fluorescence intensities were detected by Synergy H1 (BioTek). The tested strains expressed PcysIJH only (pTRG+pUC19- PcysIJH -eGFP), and co-expressed both Cbl and PcysIJH (pTRG-Cbl+pUC19- PcysIJH -eGFP), respectively. Values represented means ± SD (n = 3). *p < 0.01 was determined by one-way ANOVA and Tukey post-test.

Cbl promotes the transcription of cysH

The amino acid sequence of cbl gene was highly homologous to the cysB family, and CysB was proved to binding with the promoter of sulfur reductase (CysJIH) as a transcription factor for regulation. Therefore, it was speculated that cbl regulated the transcription of genes such as cysH, cysJ and cysI. The relative expression of cysH decreased significantly compared WT with cbl deletion by RT-qPCR, while those of cysI, cysJ revealed no differences (Fig. 3C).

Furthermore, the fusion of the promoter cysJIH (PcysJIH) and eGFP was constructed as the indicator plasmid for the fluorescent measurement. When co-expressed with Cbl, the fluorescence value was extremely significantly higher than that of the strain containing only PcysJIH (Fig. 3D). Collectively, Cbl had a positive regulation on PcysJIH.

Cbl regulates downstream cysH by binding to the P cysJIH

To confirm whether Cbl bound to PcysJIH, the PcysJIH promoter sequence and Cbl coding sequence were cloned into pBXcmT and pTGR plasmids respectively, and then co-transformed into E. coli XL 1-Blue MRF’ reporter strain for bacterial one-hybrid experiment. Only the co-expressed strain and the positive control grew on the minimum medium supplemented with six mM 3-AT and streptomycin (Fig. 4A), suggesting that the strong interaction between PcysJIH and Cbl.

Figure 4 Cbl regulated downstream cysH by binding to the PcysJIH.

(A) Results of bacterial one-hybrid. (B) Electrophoretic mobility shift assay (EMSA) for Cbl binding with PcysJIH. The 25 nM FAM-labeled PcysJIH was incubated with the increased amounts of Cbl protein. Cbl protein was titrated to the concentration of 0, 10, 20, 30, 40, 50 and 60 μM. (C) Electrophoretic mobility shift assay (EMSA) for Cbl binding with the varied size of PcysJIH. PcysJIH contained 226 bp upstream of transcriptional initiation site, PcysJIH150contained 150bp upstream of transcriptional initiation site, and PcysJIH50 contained 50 bp upstream of transcriptional initiation site. The 25 nM FAM-labeled probe DNA was incubated with 60 μM Cbl protein. The experiments were repeated in triplicate.

Next, PcysJIH was labelled with Fluorophore 6-carboxy-fluorescein (6-FAM) for electrophoretic mobility shift assay (EMSA). The Cbl protein reduced the mobility of the 6-FAM-PcysJIH DNA probe corresponding to the increased Cbl concentration with the enhanced Cbl–DNA complex (Fig. 4B). So Cbl protein was able to bind with PcysJIH following with the regulation of H2S production.

Determination of the binding region of P cysJIH with Cbl protein

To confirm the binding region of PcysJIH with Cbl protein, we truncated the full length of PcysJIH to 150 bp and 50 bp upstream of transcriptional initiation which were named as PcysJIH150 and PcysJIH50. PcysJIH150 was able to form a complex with Cbl protein, while PcysJIH50 lost the binding ability (Fig. 4C). The result suggested that the regions between 50 bp and 150 bp upstream of transcriptional initiation in PcysJIH were responsible for the binding of Cbl.

Discussion

SmpB protein is involved in the regulation of multiple biological processes such as protein invasion, bacterial movement, central metabolism, lipopolysaccharide biosynthesis, two-component system, fatty acid metabolism, high temperature tolerance, cell cycle, and stress response (Shin & Price, 2007; Ansong et al., 2009; Barends et al., 2010). And the destruction of SmpB reduces the tolerance and adaptability of bacteria (Ansong et al., 2009).

Bacterial H2S has been proved toresist oxidative stress by reacting with reactive oxygen species (ROS), H2O2, etc. or stimulate catalase and superoxide dismutase to scavenging free radicals (Kimura, 2014; Mironov et al., 2017). Besides, the oxidative stress effect of H2S is also related to the defense of bacteria against antibiotics, because many antibiotics also trigger the production of ROS when they function as the targeted inhibition (Kohanski et al., 2007). So, the effect of H2S in scavenging ROS can make it more resistant to antibiotics.

In our study, the smpB deletion of Aeromonas veronii C4 was significantly higher in H2S production than wild type under M9 culture condition (Fig. 2B), implying that SmpB deficiency enhanced the H2S generation. Indeed, smpB deletion up-regulated multiple genes in the sulfate assimilation pathway, including cysN, cysD, cysC, cysH, cysJ, cysI and cbl (Figs. 1B and 1C).

In Salmonella Typhimurium, the promoter of cysJIH (PcysJIH) is regulated by CysB (Álvarez et al., 2015), which is homologous with Cbl (Kertesz, 2000). Therefore, we presumed that Cbl was responsible for the regulation of cysH, cysJ and cysI in Aeromonas veronii C4. Cbl bound to PcysJIH and positively regulated the transcription of cysH (Fig. 3D and Figs. 4A–4C).

Previously smpB deletion exhibits more tolerance to aminoglycosides antibiotic and oxidative stress under M9 culture (Fig. 2C) (Liu et al., 2018; Wang et al., 2019). In summary, we proposed that Cbl-regulated H2S generation compensated for the resistance and survival of SmpB damage under nutrient deficiencies, contributing to the adaptation and evolution of Aeromonas veronii against extreme environment.

Conclusions

This study provided the first demonstration for the regulation between Cbl and cysJIH, and innovatively proposed the mechanism of Cbl-mediated H2S synthesis. Previously the strain of smpB deletion was observed to survive better than WT under the appropriate concentration of H2O2. In view of the function of H2S in oxidative resistance, we speculated that the accumulation of H2S increased the tolerance of oxidative resistance in SmpB deficiency. The results expanded the function of Cbl in pathogenic bacteria, and systematically explained the dynamic role of H2S in protecting bacteria from oxidative stress. These findings provide potential drug targets for aquatic diseases, offers theoretical basis for better understanding of bacterial pathogens resistance to environmental stress and supplies new ideas for clinical prevention and control of bacterial pathogens.

Supplemental Information

Supplemental Information 1 The information about the strains, plasmids and primers used in this study.

Click here for additional data file.

Supplemental Information 2 RNA-sequencing of smpB-dependent gene expression in A. Veronii C4.

Click here for additional data file.

Supplemental Information 3 Raw Data of qRT-PCR, H2S detection and Fluorescence detection.

Click here for additional data file.

Additional Information and Declarations

Competing Interests

Author Contributions

Data Availability

The authors declare that they have no competing interests.

Yidong Zhang conceived and designed the experiments, performed the experiments, analyzed the data, prepared figures and/or tables, authored or reviewed drafts of the paper, and approved the final draft.

Zebin Liu conceived and designed the experiments, performed the experiments, analyzed the data, prepared figures and/or tables, and approved the final draft.

Yanqiong Tang conceived and designed the experiments, authored or reviewed drafts of the paper, and approved the final draft.

Xiang Ma conceived and designed the experiments, authored or reviewed drafts of the paper, and approved the final draft.

Hongqian Tang conceived and designed the experiments, authored or reviewed drafts of the paper, and approved the final draft.

Hong Li conceived and designed the experiments, authored or reviewed drafts of the paper, and approved the final draft.

Zhu Liu conceived and designed the experiments, authored or reviewed drafts of the paper, and approved the final draft.

The following information was supplied regarding data availability:

The raw measurements are available in the Supplemental File and at NCBI: PRJNA493739.

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
