# Peer review of "Cbl upregulates cysH for hydrogen sulfide production in Aeromonas veronii"

_PeerJ, doi:10.7717/peerj.12058_

## Round 0.1 · original submission · Major Revisions

The reviewers found the experimental methods and results to be of high quality, but the manuscript requires thorough editing for structure, clarity of reporting results, background information and context, appropriate conclusions, and English grammar. Therefore, I am requesting a Major Revision even though I do not think that any additional experiments or analyses are required. Please respond to all of the reviewers' comments.

Note that reviewer 3 has provided the attached annotated PDF that highlights a few errors.

Reviewer 1 ·

Basic reporting

Although the manuscript is generally written in an understandable way I strongly recommend re-reading of the entire text (e.g. by someone other than the Authors, preferably a native English speaker) to help point out and improve the weaker sentences (and some grammar mistakes) that are used throughout the manuscript, and which make understanding of some ideas unnecessarily difficult. For example, some of the sentences are constructed in a too simplified way, to the point that they are lacking the crucial words: (line 68) “Hydrogen Sulfide (H2S) is an unpleasant smell with toxicity (Lindenmann et al., 2010).” I would expect the Authors to describe H2S as e.g. “gas”, “compound” etc.. Other examples: “the total amount of 1 OD600 cells were harvested”, “H2S is able to resist oxidative stress”, “many antibiotics also trigger the production of ROS when they targeting inhibit their targets”, etc. Such sentences should therefore be expanded to clarify the meaning, increase the accuracy, and make the text easier to read (fluency).

Moreover, in the example with H2S, in the aqueous environment (depending on the pH) HS- species may also be found [e.g., Hydrogen sulfide (H2S) is readily water soluble, and, at physiological pH, about two-thirds exists as hydrogen sulfide ion (HS—) and one-third as undissociated H2S; Reiffenstein et al. (1992).]. I would expect the Authors to provide more detail on the chemistry of H2S/HS- in the cells/culture fluids/culture gases, as this is vital to the understanding of the methodology used in the study. The potential presence of H2S/HS- and analysis of their concentration both as gas and in the aqueous phase should also be reflected in the Materials&Methods section.
The Authors provide some data on cys operon regulation based on the Salmonella typhimurium example (lines 73-74). Is there any data on Aeromonas species regarding this matter? If not, I would encourage Authors to indicate that, as this is likely to increase the novelty of the study.

Dividing the information provided in the Introduction section into (more) paragraphs is likely to be beneficial. For example, it is not clear whether the information about the function of Clb relates only to Salmonella typhimurium or it is a more general fact. If it is the latter case, the information about Clb should be provided in a new paragraph.

Figure 2 B - negative values on y axis (concentration) should not be present

Figure 4 A – it is difficult to observe the differences in microbial growth between the plates (low resolution/contrast)

Experimental design

Generally, in the Materials and Methods section, please add information about the number of replicates and the controls used in each of the experiments.

“LB/M9 media” - although these are typical growth media, I suggest providing more detail on their exact composition, as this often differs between manufacturers.

As mentioned above - regarding H2S detection - there is always the problem of HS- remaining in the medium. This very much depends on the pH and as such, the pH should be monitored at all times (was it?).

Were the Pb(Ac)2 paper strips submerged in the culture for 48 h, or were they wetted with the culture fluids after 48 h? Paper strip-tests tend to be more qualitative or semi-quantitative than other analytical tests. Please provide more detail about the strips used, including clear information about interpretation of the results (especially about quantitative comparison of the H2S/HS- yields based on color changes).

More details about the Cbl protein complementation procedure should also be provided in the Materials&Methods section.

Tukey’s test (named in the Figures) should be mentioned in the statistical analysis section.

Validity of the findings

“the absence of SmpB enhanced the synthesis of H2S”
Have the Authors performed any experiments to further explain this phenomenon?
Potentially, SmpB is a repressor of the cys operon, which helps to limit the production of potentially toxic H2S. H2S may be a byproduct produced at stationary phase, due to the lack of down-regulation provided by SmpB. Although this is quite crucial, this matter is only vaguely explained.

Is cysB transcription also up-regulated as a result of smpB deletion? There is no information about the transcription level of this crucial gene in the text or Figures.

The conclusions should be more focused on the outcomes of the study, and the parts about pathogenic strains and antibiotic resistance, which are not directly the object of the study should be a part of the discussion.

Additional comments

The title is a little misleading, as the Cbl affects H2S (as presented in Figure 3) regardless of SmpB presence/deletion. Of course the deletion of SmpB is a special case in which the role of Cbl may be more prominent.

Reviewer 2 ·

Basic reporting

Abstract: Line 39- re-write second messager to second messenger.

Introduction: Hydrogen Sulfide (H2S) is an unpleasant smell with toxicity (Lindenmann et al., 2010). Rewrite to Hydrogen Sulfide (H2S) is a gaseous molecule with unpleasant smell with toxicity (Lindenmann et al., 2010). 


The Introduction part should be improvised with adequate facts on how the potential findings of the study support as dug targets for aquatic diseases, since the conclusion delivers such a statement.

Lines 66-67- In all, SmpB is essential for………adaptability to stress. To which organism you mention this to? Please re-write the sentence.
Discussion: Lines 234, 237- Staring a sentence with Really or Logical is not apt for a manuscript.
The manuscript needs language editing and require technical typo corrections like space (like line 82 bothcbl instad of both cbl; line 132, 7.Electrophoretic mobility shift assay-space)

Experimental design

Materials and method:
Line 128, under subheading 6- Please describe the procedure (Bykowski et al., 2002) briefly.

Experimental design is satisfactory with required methods.

Validity of the findings

Satisfactory

Additional comments

The manuscript need minor revisions stated above.

Reviewer 3 ·

Basic reporting

no comment

Experimental design

no comment

Validity of the findings

The role of H2S in stress tolerance might be explained. Genes required for H2S synthesis might be explained in detail. The statistical significance of P<0.01 found to be less significance. If we get P<0.001 or P<0.005 more satisfactory.

Additional comments

More explanation regarding stress tolerance might be encouraged.

Annotated reviews are not available for download in order to protect the identity of reviewers who chose to remain anonymous.

---

## Round 0.2 · Minor Revisions

The manuscript is significantly improved, and the review below suggests some further changes that I agree would help to improve the final product. Please consider all of the reviewer's suggestions carefully, and you may want to seek additional editing help for grammar and clarity.

Reviewer 1 ·

Basic reporting

The manuscript has been thoroughly improved. However re-reading of the improved manuscript by a native English speaker is still recommended.

Some of the remaining minor mistakes are listed below:
line 69/70 "that contribute to reduce oxidative state" (meaning unclear, English revision needed)
line 70 "hydrogen Sulfide" (S->s)
line 71 "H2S is a gaseous molecule with unpleasant smell with toxicity" (improve the "with toxicity" part; e.g. which at high concentrations is toxic to most living organisms/live cells etc.)
line 116 "as the water solubility of H2S. (?)" (unfinished sentence?)
line 122 "there is a little evidence" or rather "there is little evidence" (confirm, as these phrases have a slight difference in meaning)
line 122 "evidence regarding Cbl regulates to cysJIH in Aeromonas species" (English revision needed) --> e.g. /regarding the role of Cbl in regulation .../
line 138 "when facing to the adverse stress" (English revision needed)
line 193 "The production of H2S were increased " -> was
line 273 "Previous smpB deletion was reported to survives better than WT " (meaning unclear, English revision needed)
line 227 "explained the the "
line 276 "to compensate for tolerance defects caused by SmpB deficiency. " (unfinished sentence)

Experimental design

The M&M section has been considerably improved.
Some minor issues are listed below:
line 148 "contained 20% 5M9 salts, " (unclear notation, it would be better to list the salts or provide a suitable reference)
line 150 "was amplified by PCR and inserted" (too vague, please add the conditions for the reactions)
line 154 "The supernatant was collected by centrifuge " (lab jargon)

Validity of the findings

It is rather uncommon to provide literature references in the conclusions section (Line 274).

Additional comments

The improved title corresponds better to the findings presented in the manuscript.

---

## Round 0.3 · accepted · Accept

Thank you for responding to all of the reviewers' concerns.